# Copper binding leads to increased dynamics in the regulatory N-terminal domain of full-length human copper transporter ATP7B

**Fredrik Orädd**[1], **Jonas Hyld Steffen**[2], **Pontus Gourdon**[2,3], **Magnus Andersson**[1]*

**1** Department of Chemistry, Umeå University, Umeå, Sweden, **2** Department of Biomedical Sciences, University of Copenhagen, Copenhagen, Denmark, **3** Department of Experimental Medical Science, Lund University, Lund, Sweden

* magnus.p.andersson@umu.se

**Data Availability Statement:** Starting structures, GROMACS topology files and trajectories are available at: https://doi.org/10.5281/zenodo.

## Abstract

ATP7B is a human copper-transporting $P_{1B}$-type ATPase that is involved in copper homeostasis and resistance to platinum drugs in cancer cells. ATP7B consists of a copper-transporting core and a regulatory N-terminal tail that contains six metal-binding domains (MBD1-6) connected by linker regions. The MBDs can bind copper, which changes the dynamics of the regulatory domain and activates the protein, but the underlying mechanism remains unknown. To identify possible copper-specific structural dynamics involved in transport regulation, we constructed a model of ATP7B spanning the N-terminal tail and core catalytic domains and performed molecular dynamics (MD) simulations with (holo) and without (apo) copper ions bound to the MBDs. In the holo protein, MBD2, MBD3 and MBD5 showed enhanced mobilities, which resulted in a more extended N-terminal regulatory region. The observed separation of MBD2 and MBD3 from the core protein supports a mechanism where copper binding activates the ATP7B protein by reducing interactions among MBD1-3 and between MBD1-3 and the core protein. We also observed an increased interaction between MBD5 and the core protein that brought the copper-binding site of MBD5 closer to the high-affinity internal copper-binding site in the core protein. The simulation results assign specific, mechanistic roles to the metal-binding domains involved in ATP7B regulation that are testable in experimental settings.

## Author summary

Living organisms depend upon active transport against gradients across biological membranes for survival. Such transport can be accomplished by ATP-dependent membrane protein transporters for which the activity must be regulated to maintain optimal concentrations in the cellular compartments. The regulatory mechanisms often involve structural responses inherent to the protein structure, which because of their dynamic nature can be hard to assess experimentally. A prime example is regulation of cellular copper levels by a copper-binding tail in the human copper transporter ATP7B. Dysregulation can cause severe diseases, for example the copper metabolism disorder Wilson's disease, which is

6412103. All other relevant data are within the manuscript and its Supporting Information files.

**Funding:** This study was supported by the Lundbeck (https://lundbeckfonden.com/en) (R313-2019-774) and the Crafoord (https://www.crafoord.se/en/) (20200739, 20180652 & 20170818) Foundations, respectively, to P.G, and the Swedish Research Council (https://www.vr.se/english.html) grant 2020-03840 to M.A. The funders had no role in study design, data collection and analysis, decision to publish, or preparation of the manuscript.

**Competing interests:** The authors have declared that no competing interests exist.

caused by mutations in ATP7B regulation machinery. Due to the practical difficulties in working with membrane proteins, most studies of ATP7B have been conducted in the absence of the membrane-bound protein core. Here, we used computer simulations of full-length ATP7B to study how structural dynamics in the regulatory tail differ between copper-bound and copper-free states. Copper induced increased dynamics in the tail, resulting in an overall movement towards the ion-binding site in the protein core. The simulations identified several, hitherto not reported, interactions between the regulatory tail and the protein core that can be targeted experimentally to enhance our understanding of this medically relevant regulatory mechanism.

## Introduction

P-type ATPase proteins transport ions across biological membranes to provide critical cofactors and uphold gradients [1]. To ensure cellular homeostasis, these ion transporters have evolved intricate regulation mechanisms that involve small peptides, lipid components in the membrane and internal protein domains [2–4]. A prime example of P-type ATPase regulation is maintenance of cellular copper levels. Copper is an essential metal that is under tight control by chaperones and membrane transporters due to its high toxicity in the free form [5,6]. The copper-transporting ATPase 2 (ATP7B) plays a key role in the cellular copper homeostasis in humans and is primarily expressed in the liver and brain [7]. At normal copper concentrations, ATP7B resides in the trans-golgi network and transports copper into the secretory pathway for incorporation into proteins, but at elevated cellular copper levels ATP7B moves to the cell membrane to export copper from the cell [7,8]. Mutations in ATP7B lead to Wilson's disease, which is characterized by copper accumulation in the liver and brain that can result in potentially fatal hepatoneurological symptoms if left untreated [9]. ATP7B is also upregulated in several cancer forms and is involved in platinum chemotherapy resistance [10]. An internal copper-binding domain is central to the regulation of ATP7B activity, but the molecular basis of regulation is still elusive.

The ATP7B transporter belongs to the $P_{IB}$ subclass of P-type ATPases and uses energy from ATP to drive copper transport. In the general Post-Albers reaction cycle, P-type ATPase proteins pass through inward-facing E1 structural states with high substrate affinity and outward-facing E2 states with low substrate affinity to achieve membrane transport [1]. The overall structure of P-type ATPases share a common topology consisting of a transmembrane (M) domain and cytosolic actuator- (A), phosphorylation- (P), and nucleotide-binding- (N) domains [1] (Fig 1A), and this topology has also been observed in crystal structures of a bacterial copper ATPase from *Legionella pneumophila* (LpCopA) [11,12] and the recent cryo-EM structure of *Xenopus tropicalis* ATP7B [13]. While there is no high-resolution structure of full-length human ATP7B, the solution structures of the A and N domains have been solved experimentally [14,15], and the crystal structure of LpCopA has enabled homology modeling of the ATP7B core domains [16,17], giving results very similar to the recent *X. tropicalis* structure [13]. The ATP7B protein also hosts a large regulatory N-terminal tail that contains six independently folded metal-binding domains (MBD1-6) connected by long, unstructured linkers [18] (Fig 1A). The solution structures of the MBDs have revealed that they share the same ferredoxin-like fold with one copper-binding CXXC motif each [19–23]. Nanobody binding, nuclear magnetic resonance (NMR) relaxation, and electron paramagnetic resonance (EPR) spectroscopy have showed that the N-terminal tail is organized into two functional units composed of MBD1-3 and MBD5-6, with MBD4 either belonging to the MBD5-6 complex or

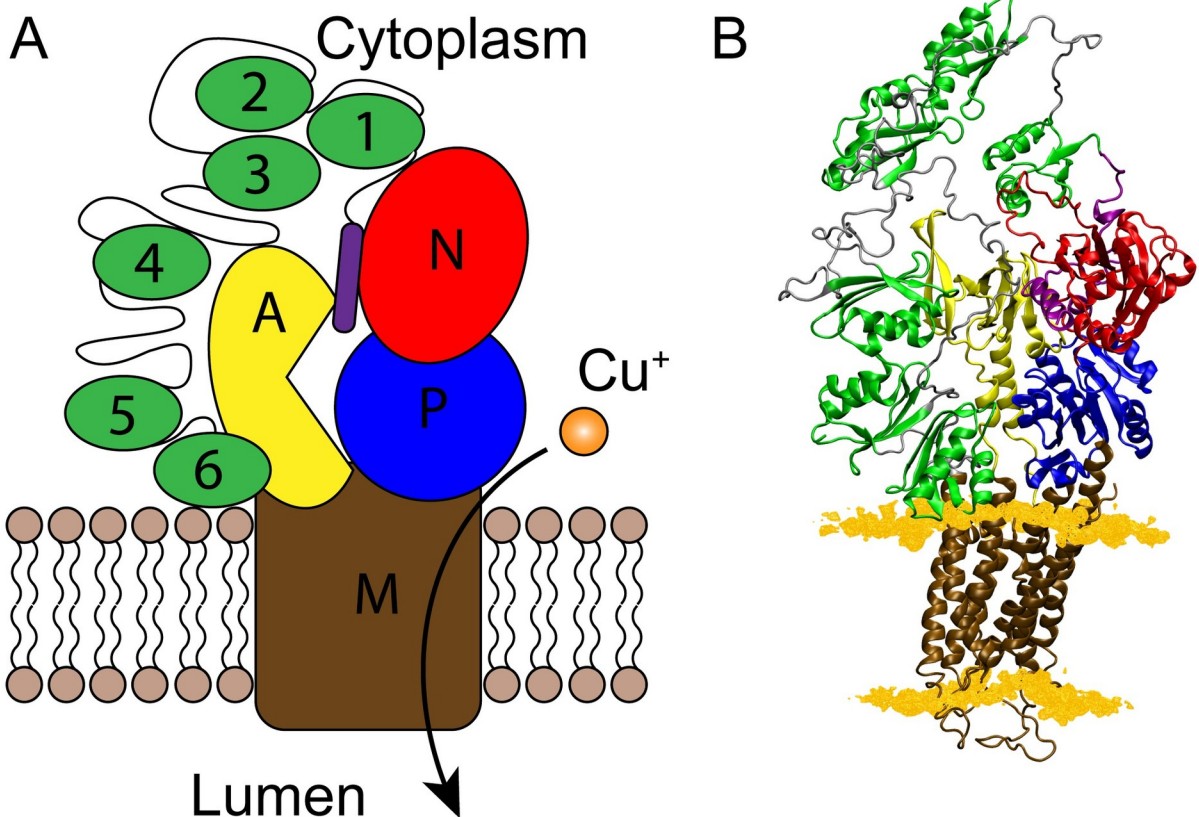

**Fig 1. Structure of the ATP7B protein core and metal-binding domains. (A)** Schematic and **(B)** all-atom homology model of the human copper ATPase ATP7B in the E2.P$_i$ state. The N-terminal peptide is shown in purple, MBD1-6 in green, the A domain in yellow, the P domain in blue, the N domain in red, and the M domain in brown. The lipid headgroups are shown as light brown spheres (A) and orange surface (B).

serving as a linker between the two groups [19,24,25], and reviewed in [26]. There is also an unstructured N-terminal peptide preceding MBD1 in the sequence that contains a signal peptide for trafficking of ATP7B to the cell membrane [27,28].

Positioning of the MBDs relative to the core protein is still uncertain, especially for MBD1-4 which have not been observed in any crystallography or electron microscopy data. Although electron densities have been resolved for MBD5 and MBD6, or corresponding MBDs in homologs with fewer MBDs, their positions vary greatly and the density is often vague. The MBD5 and MBD6 positions were resolved in the cryo-EM structure of a deletion mutant of *X. tropicalis* ATP7B lacking MBD1-4 (ATP7B-ΔMBD1-4) in the absence of copper, where they primarily contacted the A and P domains, with MBD5 placed above the copper entry site in the M domain [13]. However, a significant fraction of the particles lacked density for MBD5 and MBD6, suggesting that the observed positions were not the only ones occupied in the apo state [13]. Low-resolution electron densities in *Archaeoglobus fulgidus* CopA and human ATP7B-ΔMBD1-4 suggested that the MBDs closest to the core protein in the sequence are located on the opposite side of the A domain compared to the *X. tropicalis* structure, and the electron density in human ATP7B placed one MBD closer to the membrane and the copper entry site [29–31]. In all the suggested positions, the MBD(s) would have contact with the A, P and N domains, which is in agreement with co-purification experiments that identified interactions between the ATP7B N-terminal tail and the A domain [32] and the P and N domains [33]. In addition, NMR chemical shift perturbation experiments showed interaction between the N-

terminal peptide and the N domain, and it has been suggested that the N-terminal peptide binds at the interface between the A and N domains [34]. Similar interfaces exist in P4- and P5-ATPases, which have autoinhibitory tails that bind at the interface of the A, N and P domains [35,36].

Copper is delivered to the ATP7B N-terminal tail by the copper chaperone Atox1 [7,37], which has also been suggested to deliver copper to the intramembrane copper binding site [38,39]. Copper binding to the N-terminal tail leads to both limited local structural changes in individual MBDs [18] and changes in the dynamics of the entire N-terminal tail [34,40,41]. For instance, MD simulations showed that copper binding leads to reduced root mean square fluctuations (RMSF) and subtle movements in amino acid side chains in single MBDs [42]. In MD simulations of two-domain MBD constructs, copper binding to one MBD induced correlated changes in Cα fluctuations in the other MBD [43]. Moreover, mutations introduced in the CXXC motif, most notably in MBD2 and MBD3, were observed to change the ability of other MBDs to bind copper, likely via changes to hydrogen bonding networks involving the cysteine thiol groups [44]. The subtle local effects of copper binding alters the dynamics of the entire N-terminal tail and results in protein activation [23]. Such changes in structural dynamics are difficult to monitor, but copper binding has been reported to cause increased mobility in the MBD1-3 part of a MBD1-6 construct [24,34], and result in a more compact formation in a MBD1-4 construct [41]. However, the underlying mechanism of copper-dependent changes in structural dynamics is currently unknown.

The N-terminal tail is generally believed to be autoinhibitory, with copper binding leading to a release of this inhibition [45,46]. Several inhibition mechanisms have been proposed, involving both the MBD1-3 and MBD5-6 groups. The MBD5-6 complex is important for copper transport, and deletion or mutation of both MBD5 and MBD6 renders ATP7B inactive [4,47–49]. This is similar to prokaryotic CopA proteins with one or two MBDs, where deletion or mutation impairs activity [46]. For single-MBD CopA proteins it has been suggested that the copper-free MBD interacts with the A domain to prevent domain movements [50] or that the MBD blocks access to the membrane ion-binding site [46]. In ATP7B, the situation is more complex due to the presence of six MBDs. The MBD5-6 complex has been proposed to play a similar role as in single-MBD prokaryotes [46], but there is also evidence suggesting that MBD5 or MBD6 are directly involved in copper transfer to the intramembrane copper binding site [47]. The MBD1-3 complex, however, is believed to be purely regulatory since it can be mutated or deleted without much effect on total activity [4,34,47]. One proposed mechanism is that the copper-free MBD1-3 complex is disrupted upon delivery of copper by Atox1 chaperones, which is then followed by enhanced MBD1-3 dynamics [34,47]. The movements of MBD1-3 could then enable the N-terminal peptide to dissociate from the interface between the A and N domains, thereby enabling ATP hydrolysis to trigger ATP7B ion transport [34]. Such a mechanism is in agreement with co-purification data showing that copper binding weakens the interaction between the N-terminal tail and a P-N domain construct and increases the ATP affinity of the P-N construct [33]. The mechanism is also similar to the autoinhibitory regulation mechanisms of P4- and P5-ATPases, where a regulatory terminal tail binds at the interface of the cytosolic domains, and upon a stimulatory signal the tail is removed from its binding site, thereby releasing the autoinhibition [35,36]. Despite recent progress in understanding how the isolated ATP7B N-terminal respond to copper binding, the changes in N-terminal structural dynamics in the presence of the protein core domains are still elusive.

In this work, we constructed a homology model of the human ATP7B core domains and N-terminal tail based on the LpCopA crystal structure, solution structures of the A and N domains, NMR structures of the six MBDs, and current information about domain

interactions and performed MD simulations to study copper-induced differences in the dynamics of the protein N-terminal tail. We found the N-terminal tail, in particular MBD2 and MBD3, to show higher mobility and to be more distant to the core domains upon copper binding. We also identified a copper-dependent interaction between MBD5 and the linkers connecting the A- and M-domains that positioned the MBD5 copper-binding motif closer to the ion-entry site in the M domain. Thus, the simulation data are congruent with an activation mechanism where MBD1-3 act as a switch that lose inter-domain interactions upon copper binding and results in priming the structure for copper delivery to the internal transport sites via MBD5/6. The simulation data predict several putative points of regulatory interactions that are testable in experimental settings to further improve understanding of ATP7B regulation.

## Results

To identify copper-dependent structural dynamics in ATP7B regulation, we performed multi-microsecond MD simulations of an ATP7B homology model with (holo) and without (apo) copper bound to the six MBDs. The homology model was obtained from combining structural information from the bacterial homolog core protein, individual metal-binding ATP7B domains, NMR-guided modeling of N-terminal peptide, and modeling of the linker domain. The initial relative positioning of the metal-binding domains was obtained from experimental studies (see the Methods section for details). A replica exchange protocol was used to enhance the sampling and to avoid simulations from becoming trapped in local energy minima. We first initiated six independent replicas per copper-binding state from the same conformation of the homology model inserted into a DOPC membrane (reA_apo and reA_holo). These two sets of replica exchange simulations were followed by two second sets of replica exchange simulations (reB_apo and reB_holo) that were initiated from unique starting conformations obtained from the reA simulations (S1 Fig). The simulation data were analyzed as independent trajectories and statistical tests were used to determine significant differences between the apo and holo states. The combined simulated dynamics showed that the N-terminal tail explored a wide conformational space, and that several significant differences existed between the apo and holo states.

### Bound copper ions induce a more extended N-terminal tail

To visualize differences in N-terminal tail dynamics, we constructed free energy surfaces for the apo and holo states (reA simulations), defined by the N-terminal tail radius of gyration and root mean square deviation (RMSD) with respect to the starting structure (S1A and S1B Fig). We found that relatively compact conformations dominated both the apo and the holo state, but that the holo state also sampled more extended conformations. To explore the possibility of either the holo or apo simulations being trapped in local minima, we initiated a new set of replica exchange simulations (reB) starting from frames in the reA simulations that displayed more extended conformations of the N-terminal tail. Similar free energy surfaces were then constructed from the replica simulations started from extended conformations (S1C and S1D Fig). These reB free energy surfaces showed that the apo state quickly moved back to compact conformations with similar radius-of-gyration as in the reA run. In contrast, the reB holo state remained extended for the duration of the simulations. The combined free energy surfaces for the reA and reB apo state simulations showed that local energy minima were concentrated to a radius-of-gyration below 3.6 nm (Fig 2A). While the holo state reA and reB simulations also visited more compact states, several minima in the free energy surface corresponded to a radius-of-gyration above 4 nm (Fig 2B). To find which holo–apo differences were statistically significant, we calculated differences in the distance between the center-of-

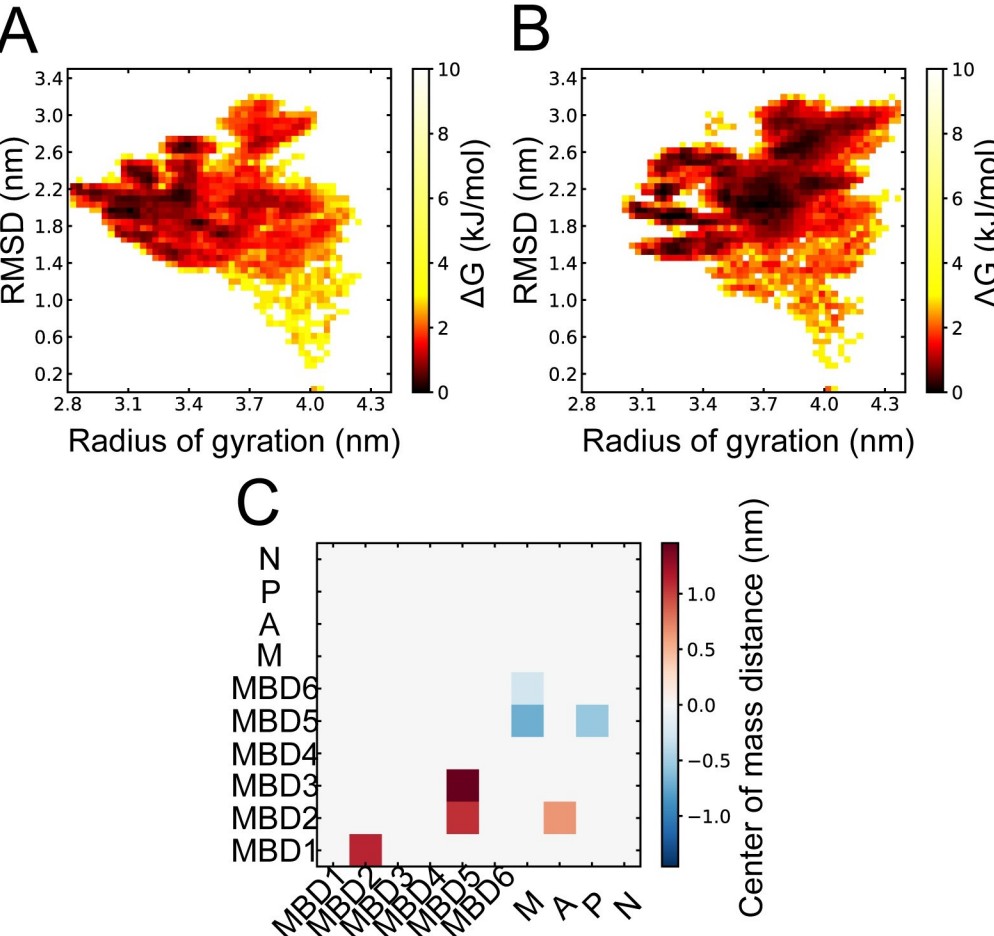

**Fig 2. Overview of the conformational differences in the ATP7B N-terminal tail.** Combined free energy surfaces from the reA and reB simulation sets defined by radius of gyration and RMSD to the starting structure in the (**A**) apo and (**B**) holo states. Darker shades correspond to higher sampling frequency. (**C**) Statistically significant ($p < 0.05$) differences in average domain-domain center-of-mass distances between holo and apo state simulations calculated as absolute distance (holo)–absolute distance (apo) for MBD1-6, and the M, A, P, and N protein core domains.

mass of the individual MBD1-6 and also to the protein core domains M, A, P, and N (Fig 2C). Significant differences ($p < 0.05$) were obtained between MBD2, MBD3, and MBD5. In the holo state, MBD2 and MBD3 were more distant to MBD5, and MBD2 was also more distant to MBD1 and the A domain. In addition, MBD5 was closer to the M and P domains, and MBD6 was slightly closer to the M domain.

## The N-terminal tail extension stems from changes in MBD2, MBD3 and MBD5 dynamics

Because the locations of MBD2, MBD3, and MBD5 differed significantly between holo and apo simulations, we dissected their positions into x, y, and z center-of-mass coordinates to better resolve inter-domain dynamics, where x and y are in the plane of the membrane and z is perpendicular to the membrane. No statistically significant differences were found in the z dimension, but several were found in the x and y dimensions. The MBD2 and MBD3 positions differed significantly ($p_x < 0.05$) along the x coordinate and were more likely to be positioned further away from the core domains in the holo state (Fig 3A and 3B). The position of MBD5

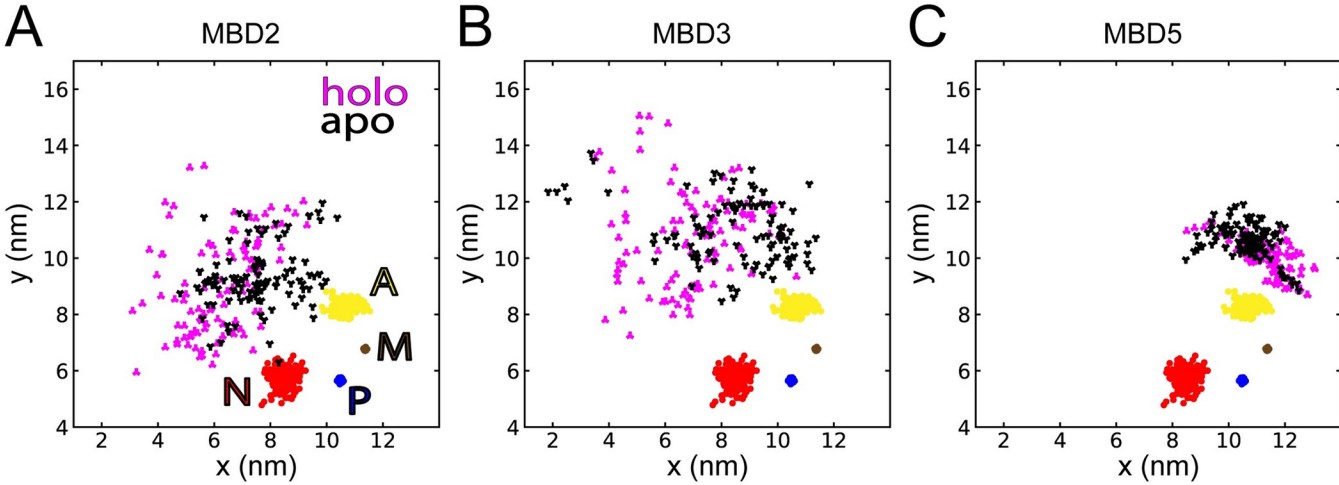

**Fig 3. Copper binding affects the dynamics of MBD2, MBD3 and MBD5.** The x- and y-coordinates of (**A**) MBD2, (**B**) MBD3, and (**C**) MBD5 in the holo (magenta) and apo (black) states with respect to the protein core domains M (brown), A (yellow), P (blue), and N (red).

was also significantly different between the two states ($p_x < 0.05$, $p_y < 0.05$), in particular relative to the A domain of the protein core, which MBD5 moves around towards the M domain in the holo state (Fig 3C).

To determine if the observed differences in domain positions were correlated with the observed extension of the N-terminal tail, we calculated cross-correlations between the radius of gyration of MBD1-3, MBD1-4, MBD1-6 and the x, y and z coordinates of the six MBDs (S2 Fig). Indeed, several significant ($p < 0.05$) correlations were found. The position of MBD3 showed the strongest correlation with the MBD1-3 radius-of-gyration, but also with the radius-of-gyration of MBD1-4 and MBD1-6. We also found equally significant, but weaker correlations between MBD2 x- and z-positions and MBD1-6 and MBD1-3 radii-of-gyration, and between the MBD5 x position and radius-of-gyration of MBD1-6. Hence, the differences in position of MBD2, MBD3, and MBD5 were correlated with extension of the N-terminal tail. In addition, the observed differences in MBD5 positions, in particular the z coordinate, were correlated with the radius-of-gyration of MBD1-3, MBD1-4 and MBD1-6. As a consequence, MBD5 was positioned closer to the membrane when the N-terminal tail was in an extended conformation. Curiously, this correlation was strongest for the MBD1-3 fragment that does not include MBD5, which implies that inter-domain dynamics in the MBD1-3 complex can affect the position MBD5 relative to the protein core and the membrane.

### Copper induces differences in N-terminal tail contact patterns

To characterize inter-domain contact patterns in the apo and holo states, we calculated 4.5 Å cut-off residue-residue contact maps. The interactions in the apo and holo states were located primarily at the A and N core domains, the N-terminal peptide, and the MBDs (Fig 4A and 4B). Within the N-terminal tail, we observed significantly more contacts involving linker regions compared to direct interaction between the MBDs (S3A Fig), in particular within MBD1-3 (S3B–S3E Fig). The high frequencies at residues 124–127 (MBD1), 140–144 and 209–214 (MBD2), and 254–258 (MBD3) were adjacent in the sequence, but high-frequency contacts also involved more central residues in the MBDs. For instance in MBD2 and MBD3 where loops opposite to the CXXC motif showed linker contacts (S3C–S3E Fig). In addition, MBD2 displayed linker contacts originating from the CXXC motif (residues 152–156) and the neighboring loop (residues 178–182) (S3D Fig).

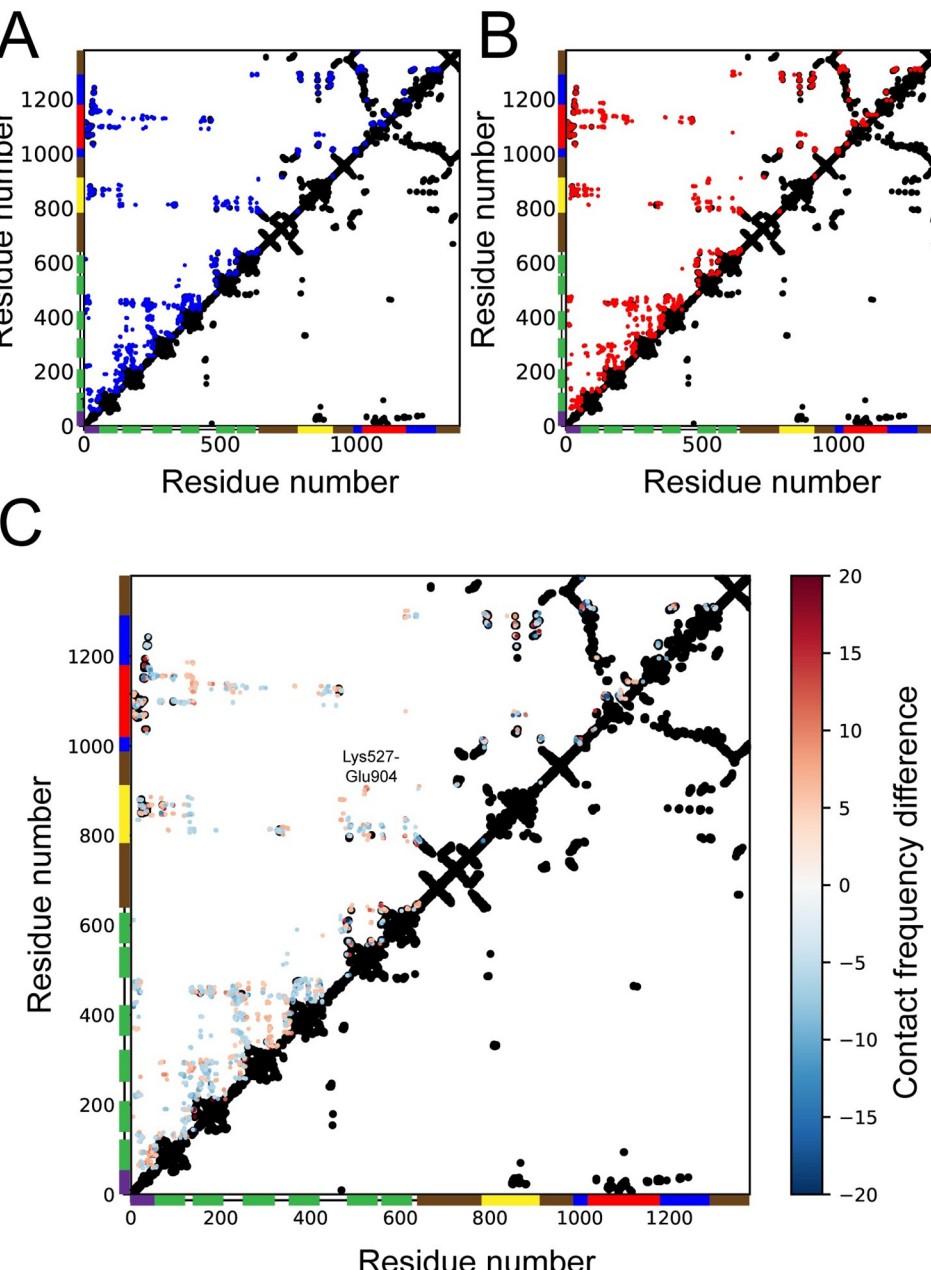

**Fig 4. Interdomain contacts in the regulatory and core domains of ATP7B.** Contact frequency maps with contacts defined as heavy atoms within 4.5 Å present in > 5% of the simulation frames of the **(A)** apo and **(B)** holo states. **(C)** Differences in residue-residue contacts between the apo and holo states (holo frequency–apo frequency) averaged over all simulation frames. The starting structure contact map (Cβ-Cβ within 8 Å) is shown in black. Domain coverage is shown as vertical and horizontal bars with the N-terminal peptide in purple, MBD1-6 in green, the M domain in brown, the A domain in yellow, the P domain in blue and the N domain in red.

To better resolve the differences, we subtracted average contact frequencies in the apo state from the holo state (Fig 4C). The resulting difference frequency plot showed several differences in contacts at the N-terminal peptide, MBD3, MBD4, MBD5, the A domain and the N domain. In the apo state, MBD5 contacted the central parts of the A domain, while in the holo state MBD5 interacted with the A-M linker regions. In addition, several contacts between the A

domain and the linker between MBD1 and MBD2 were exclusive to the apo state. Within the N-terminal tail, the holo state showed less contacts than the apo state, in particular between MBD2 and MBD3 (S4 Fig). In the apo state, the residues with the highest contact frequencies were Leu273, Glu276, Glu277, and Glu293 (MBD3) and Arg166, Lys167, Arg173, and Lys175 (MBD2). The contacting residues were largely different in the holo state. Here, the main interacting residues were Glu293, Lys300, and Thr258 (MBD3) and Gln155, Ser156, Leu178, and Ser179 (MBD2). Hence, copper binding induced a shift in the contact surface of MBD3 from the helix connected to the CXXC motif to a patch on the beta sheet opposite to the copper-binding site, and the MBD2 interaction surface with MBD3 changed from the loops opposite to the CXXC motif in the apo state to the CXXC motif and the neighboring loop in the holo state (S4C and S4D Fig). MBD3 also showed contacts with MBD4 in the holo state, albeit with low frequency, and MBD4 had fewer contacts with the MBD2-MBD3 and MBD4-MBD5 linkers in the holo state, which could reflect an enhanced mobility.

Finally, we observed differences in contact patterns between the N-terminal peptide and MBD1, the A domain and the N domain. In the holo state, the more terminal parts of the N-terminal peptide (residues 7–20) contacted A and N domain residues that were in contact with the central part of the N-terminal peptide (residues 20–35) in the apo state, and upstream residues 36–55 contacted MBD1 in holo state instead of the N domain. The contact patterns in the apo state were more similar to the starting structure, indicating that the increased dynamics in the holo state caused the N-terminal peptide to be pulled out of its binding position at the core protein.

## The MBD5 copper-binding site approaches the ion-entry site in the M domain

The simulations showed interactions between MBD5 and the A- and M-domains, in particular in-between residues Lys527 and Glu904 (Fig 4), which were also in contact in the AlphaFold [51,52] model of ATP7B. In this model MDB5 was placed next to the A domain with its copper-binding site directed towards the membrane domain copper-binding site (S5 Fig). We also compared to co-evolution contacts predicted by the EVcouplings server [53], which showed similar interactions but instead with MBD6. The contacts included Asn581-Ser726, Ser602-Lys785, Lys603-Glu904, and Leu605-Glu905, which would place MBD6 with its copper-binding site near the membrane entry site (S6 Fig). Although these contacts were not observed in the simulations, a contact was formed between Lys527, which is the MBD5 equivalent of Lys603, and Glu904.

To further characterize the interaction between MBD5 and the A domain, we determined interdomain residue-residue contacts (S7A and S7B Fig). The residues with the highest contact frequencies in MBD5 were the beta-sheet residues Lys531, Glu529, Leu520, and Lys489. The contacting residues in the 520–531 range showed ~10% higher contact frequencies in the holo state. In the A domain, three clusters of residues were found to contact MBD5. The first cluster, which also showed the highest contact frequency, contained residues Thr801 and Glu802 and contacted MBD5 in both the apo and the holo state. The second cluster, consisting of residues between Glu810 and Gln826, showed more differences between the two states, with Gln819 and Glu823 being the main interaction points in apo and holo, respectively. The third cluster was exclusive to the holo state and consisted of residues between Asn892 and Met908, which included residues Asn892, Glu904, and Glu905 that were predicted in the co-evolution analysis to interact with MBD6. The three A-domain clusters are located in linkers that connects directly to transmembrane helices two and three (cluster one and three) and in the surface-exposed beta strands in the A domain (cluster two) (S7C–S7E Fig).

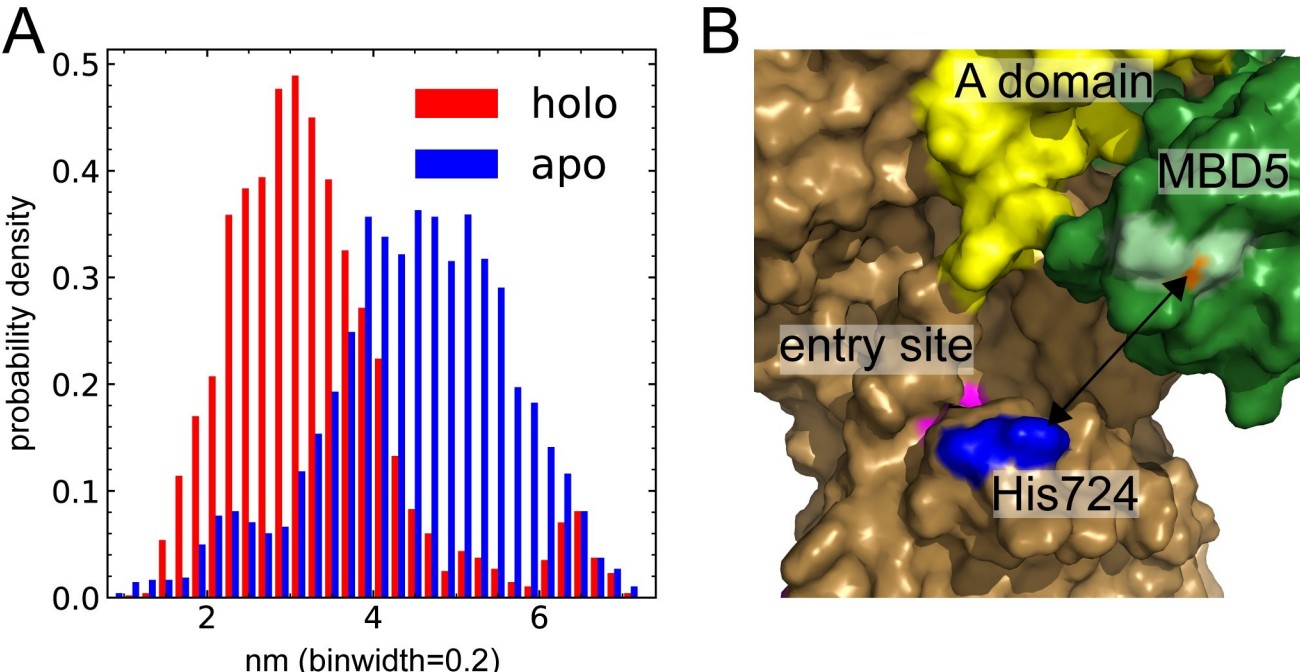

**Fig 5. MBD5 is located closer to the copper entry site in the holo state.** (A) Distance between the copper-binding sulfur atoms of MBD5 and the proposed copper ligand His724. (B) Structural view of MBD5 (green), the A domain (yellow), the copper entry site (magenta), His724 (blue), copper-binding cysteines (light green) and copper (orange) bound to MBD5 in the holo state. The rest of the protein is colored brown.

To investigate possible effects of apo-holo differences in MBD5 position and contacts to the A domain, we measured the distance between the center of mass of the MBD5 copper-binding cysteine sulfur atoms and the sidechains of residues that might serve as copper ligands in the membrane domain copper entry site [12,13,54,55]. The copper-binding motif of MBD5 was closest to His724, which has been suggested as a copper ligand based on the *X. tropicalis* cryo-EM structure [13], and it was closer in the holo state (3.3 ± 1.1 nm) than in the apo state (4.5 ± 1.1 nm) (Fig 5A). The copper-binding sulfurs were approximately 1 nm from His724 in the closest position, and in this position the copper was directed away from the entry site (Fig 5B).

## Discussion

In this work, we simulated a homology model of ATP7B including the N-terminal tail with and without copper bound to the MBDs. We found several significant differences in MBD mobility and position (Fig 6). In the copper-bound state, MBD2 and MBD3 exhibited a higher degree of mobility and occupied more distal positions compared to the other protein domains, in agreement with previously published models of ATP7B regulation [34,47]. In addition, MBD5 interacted more closely with the A domain in the holo state and tended to occupy positions closer to the M domain copper entry site. Our findings show that the N-terminal tail of ATP7B becomes more dynamic and extended when copper is bound to the MBDs and suggest a potential interaction surface for MBD5 near the ion-entry site in the M domain.

### N-terminal tail contacts to the core protein are transient and non-specific

In both the apo and holo simulations, the MBDs explore a wide configurational landscape and do not appear to click into specific positions with respect to the core protein. This is

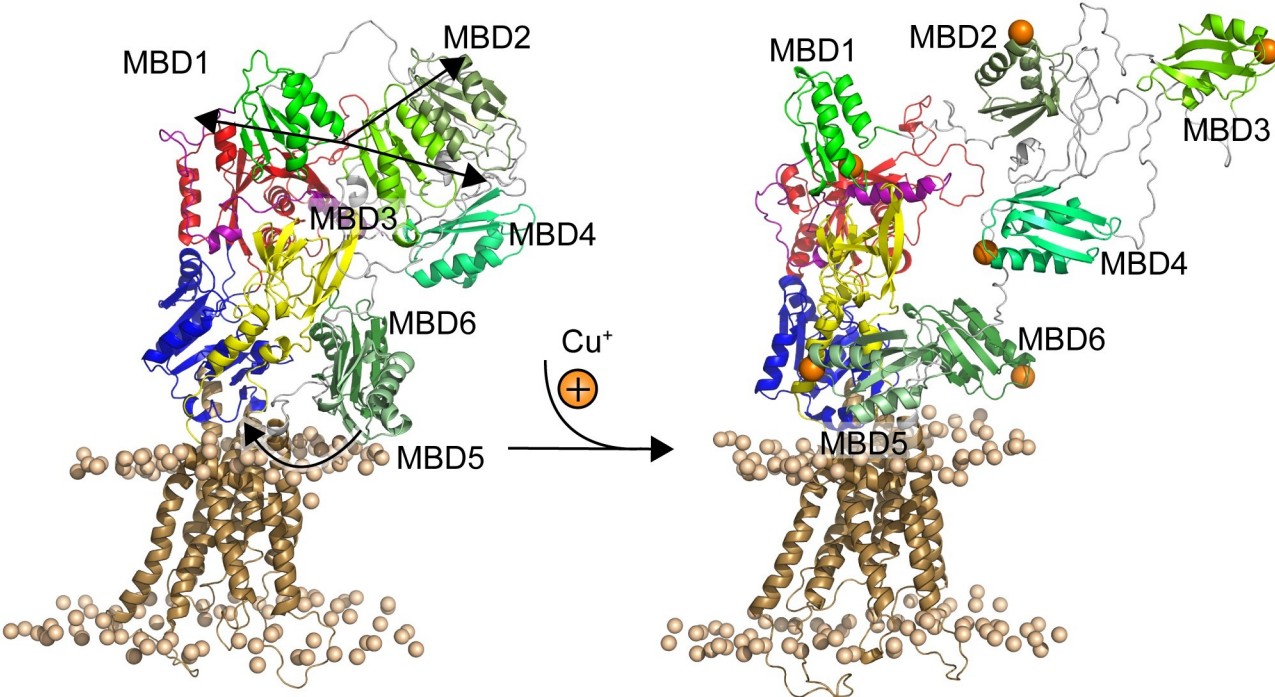

**Fig 6. Proposed ATP7B regulatory dynamics triggered by binding of copper to the six MBDs.** Representative structures from the apo and holo states with the same coloring as in Fig 1 for the core protein. The MBDs are shown in shades of green and bound copper is shown as orange spheres, and trends in MBD movements induced by copper binding are shown as black arrows. Upon binding of copper to the MBDs, the interactions between MBD1-3 decreases and MBD2-3 become more mobile. In addition, MBD5 moves closer to the copper entry site.

corroborated by coevolution analysis of ATP7B proteins with six MBDs, which showed no significant contacts between the MBDs and the core enzyme (S8 Fig). Therefore, the interactions between the MBDs and core domains are likely of a transient, non-specific nature rather than defined contacts between specific residues. In the apo state, the closeness of the N-terminal tail to the cytoplasmic A and N domains likely introduces significant crowding that might hinder functional movements. In contrast, in the holo state, the MBDs were more likely to dissociate from the core domains, thereby reducing the crowding and enabling functional domain movements and ion transport.

## MBD2 and MBD3 dynamics could trigger release of N-terminal tail autoinhibition

MBD1, MBD2 and MBD3 form a transiently interacting domain cluster [24], and it has been proposed that a reduction in interactions within this MBD1-3 domain group is a critical early step in ATP7B activation [34,47]. The copper chaperone Atox1 delivers copper to ATP7B MBDs, and NMR transverse relaxation rate experiments have shown that copper delivery by Atox1 increases the tumbling rates of individual MBDs in the MBD1-3 group, which indicates a loss of interactions within the MBD1-3 group [34]. In our simulations, we observed more contacts between MBD2 and MBD3 in the apo state, in particular surrounding the CXXC motif, which oriented the domains side-by-side (S4C Fig). In the holo state, the CXXC motif of MBD2 interacted with a beta sheet in MBD3, but with overall lower contact frequencies (S4D Fig). The observed interactions are different from those observed in previous studies of MBDs and Atox1 [34,37,43] and could therefore represent new functional interactions. In the

apo state, the copper-binding residues of MBD2 were not involved in the interaction, potentially leaving them available for copper delivery by Atox1, which has been suggested as the first step of copper delivery to ATP7B [34,56]. Delivery by Atox1 could disrupt the interaction [25] and make the MBD3 CXXC motif available for copper transfer. Binding of copper to both MBD2 and MBD3 could then cause the MBD1-3 complex to dissociate, as suggested previously [34,47]. Such a scenario could be tested with a simulation strategy involving copper bound to different combinations of MBDs, which is outside the scope of the present work. However, we observed an increased mobility for MBD2 and MBD3 in the fully copper-bound state. This was especially pronounced when the simulations started from conformations where the N-terminal tail was extended, indicating that starting from extended positions might mimic the effect of Atox1 interfering with MBD1-3 interactions upon copper delivery. Our results therefore suggest that copper bound to the MBDs prevents MBDs 1–3 from re-forming stable interactions, which maintains an extended conformation of the N-terminal tail. The observed copper-dependent MBD interaction patterns are in agreement with experiments showing a MBD1-4 construct to be more compact upon copper binding by $CuCl_2$ [41] compared to Atox1-mediated delivery to a MBD1-6 construct [34].

Increased mobility in the N-terminal tail, in particular MBD1-3, has been proposed to pull the N-terminal peptide from its interaction site between the A and N domains [34]. We observed copper-dependent differences in contact patterns around the N-terminal peptide that could represent early stages of the N-terminal peptide being pulled out of its binding site in the holo state. These differences are likely a result of the increased N-terminal tail mobility in the holo state, as Yu et.al. suggested based on their NMR data [34]. A complete removal of the peptide from the interface might require more simulation time or it might only be possible in the E1 state conformation of the core protein. In the E2 state, the peptide could be partially trapped between the A and N domains, which are more tightly packed against each other compared to in an E1 state.

## Copper-dependent MBD5 rearrangement can prime ATP7B for copper transfer

In our simulations, we observed a significant difference in MBD5 position between the apo and holo states (Fig 5A). In the holo state, MBD5 engaged in several contacts with the linkers connecting the A and M domains, which resulted in a MBD5 location closer to the copper entry site in the M domain (Fig 5B). A possible explanation is that copper binding expands the available range of positions for MBD5, allowing it to dissociate from an inhibitory binding site. However, the *X. tropicalis* structure shows that the most stable position of MBD5 in apo-ATP7B-ΔMBD1-4 is closer to the top of the P domain, away from the intramembrane copper-binding site [13]. Our observed shift in MBD5 position could therefore suggest MBD5 to gain a function in the holo state, rather than only releasing inhibition. The observed MBD5 position is similar to a proposed MBD position based on cross-linking in an *Enterococcus hirae* CopA protein [57], and it is also reminiscent of the MBD5 position predicted by AlphaFold (S5A Fig), where MBD5 was docked onto the copper-loading platform with its copper-binding site directed into the membrane copper binding site (S5B Fig). The observed MBD5-A domain contacts also corresponded to co-evolution predicted contacts between MBD6 and the A domain, which had high probability rating despite the relative lack of sequences covering the N-terminal tail. Several of the contacting residues in MBD5 and MBD6 are conserved in vertebrate ATP7Bs [16], and the surface-exposed linkers between the A domain and the TM helices are conserved across CopA proteins [11]. In the simulations, as well as in the coevolution contacts and the AlphaFold model, a conserved lysine (Lys527 in MBD5 and Lys603 in MBD6)

showed interactions to Glu904 in the A domain. Given that both AlphaFold and the EVcouplings server use coevolution to predict structures and contacts it seems likely that they picked up the same coevolution pattern but placed different MBDs in a "copper delivery position". It is therefore possible that both of these MBDs can take a delivery position, which is in agreement with observations showing that either MBD5 or MBD6 must have an intact copper-binding site for the protein to be active [47,48]. Our simulations suggest that copper binding to the N-terminal tail allows MBD5 (or, potentially, MBD6) to approach and possibly dock to the membrane copper binding site, and that this enhances protein activity. The simulations did not sample MBD5 positions as close to the copper loading platform as was predicted by AlphaFold and EVcouplings, which can possibly be inherent to the E2 state where the helices connecting the A- and M-domains partly block the putative copper delivery site. It is also possible that the movement of MBD5 is restricted by the position of MBD6, which in our simulations was located on the opposite side of the A domain. The MBD6 position observed in the recent *X. tropicalis* ATP7B E2 structure would likely restrict MBD5 less and allow it to dock to the copper-loading platform [13]. Regardless, the copper-dependent movement of MBD5 closer to the membrane copper-binding site in the E2 state could possibly prime the protein for the next round of ion transfer upon transitioning to an ion-accepting E1 state, either by copper delivery to the membrane ion-binding site or by cooperative stimulation of A domain movements.

In conclusion, we found that copper binding to the ATP7B N-terminal tail increased the mobility of MBD2 and MBD3 and caused MBD5 to move closer to the copper entry site in the membrane domain. This supports a model where MBD1-3 lose interdomain contacts and become more mobile upon copper binding, leading to a release of inhibitory interactions with the ATP7B core domains. The position of MBD5 in the holo state shows a potential interaction surface to the A domain, which based on contact predictions from evolutionary contacts could potentially also be filled by MBD6. This interaction brings MBD-bound copper closer to the ion entry site and could prime the protein for the next round of the transport cycle. Experimental validation of the predicted interaction sites and their importance for ion transport can further advance our understanding of the regulation mechanism in copper transport.

## Methods

### Initial homology model

The starting structure for the core domains was obtained from an ATP7B homology model [16] based on a crystal structure of the bacterial copper-transporting ATPase LpCopA (pdb-id 3rfu) [11], which was the closest known structure at the time of writing. The starting structures for the metal-binding domains were taken from the following protein data bank NMR structures: MBD1 (pdb-id 2N7Y) [21], MBD2 (pdb-id 2LQB) [22], MBD 3 and 4 (pdb-id 2ROP) [20], MBD 5 and 6 (pdb-id 2EW9) [19]. The starting conformation of the N-terminal peptide (residues 5–54) was modeled with the *de novo* folding server PEP-FOLD3 [58] using the NMR solution structure of the ATP7B N domain (pdb-id 2ARF) [14] as the receptor, with the interaction patch defined by the NMR contacts reported in [34]. The starting N-terminal peptide structure was chosen from the top 10 predictions based on steric compatibility with the core domains of ATP7B (after alignment of the N domain) and agreement with secondary structure prediction servers. The initial placement of MBD5-6 was as in Gourdon et al. [16]. The internal structure of MBD1-4 was obtained from [18] and placed to avoid steric clashes with the core domains and MBD5-6 upon generating the missing linker regions using SWISS-MODEL [59]. The full starting model covers residues 5–1378 of the ATP7B protein, with the N-terminal tail domain arranged into three groups; MBD1-3, MBD4, and MBD5-6, in accordance with experimental studies [23,24,34].

## Copper parameters

In the simulations of the copper-bound holo state, the copper was bound to the sulfur atoms of the copper-binding cysteines in the MBD CXXC motifs. The copper center angles, the Cu-S bond length, and associated force constants have been published [42,60], as well as the van der Waals parameters [61] and partial charges [42]. The Cα and Cβ partial charges were obtained from [60]. The copper, cysteinate, Cα and Cβ partial charges were combined with charmm36m cysteinate (CYM) charges for the remaining atoms (N, CO, H) and the total copper-center charge was modified to -1, spread over all atoms except S and Cu. The resulting set of parameters is very similar to those used in [42] and [43] for simulation of one- and two-domain constructs of the ATP7B MBDs. The resulting copper coordination geometry is linear bi-coordinated rather than tetrahedral as in [61], which is in agreement with QM/MM MD simulations on MBD3 and MBD4 [37] (S9 Fig).

## Building the systems

The apo state ATP7B homology model was inserted into a lipid bilayer consisting of 894 DOPC molecules using the bilayer builder [62] module of CHARMM-GUI [63] and solvated with TIP3P [64] water molecules. The total charge of the system was neutralized by adding 6 $K^+$ counterions. In the holo system, copper-binding cysteine residues were changed into CYM, and one $Na^+$ was added between the cysteinate S atoms in each copper-binding motif. This modified homology model was inserted into a lipid membrane with 894 DOPC molecules using the CHARMM-GUI bilayer builder [62,65] and solvated with TIP3P water molecules [64]. The charge was neutralized with 13 $K^+$ counterions. The $Na^+$ ions were then replaced with copper and the copper-binding CYM residue parameters were modified as described.

## Molecular dynamics simulations

In total, one 1 μs extended simulation and two 200 ns six-replica replica exchange simulations (reA and reB) were performed for the apo and holo states (6.8 μs total) using GROMACS 2019 [66]. For the extended and reA simulations, the systems were energy minimized with a steepest descent method until the maximum force on any atom was $< 1000$ kJ mol$^{-1}$ nm$^{-1}$. Atom positions were equilibrated in six steps, during which atom restraints were gradually released. The first two equilibration steps were run for 25 ps with constant temperature (303.15 K) and volume (NVT ensemble), with a time step of 1 fs. The next four equilibrations were run in constant temperature (303.15 K) and pressure (1 bar) (NPT ensemble), the third step for 25 ps with 1 fs time step and the last three for 100 ps with 2 fs time step. The equilibrations used a Berendsen thermostat [67] and a Berendsen semiisotropic barostat [67]. The production simulations were run with constant temperature and pressure (1 bar) using a Nose-Hoover thermostat [68,69] and a Parrinello-Rahman barostat [70,71]. The protein and lipids were described by the additive all-atom forcefields C36m Protein [72] and C36 lipids [73], respectively, and the water molecules were described with the TIP3P model [64]. The electrostatic interactions were calculated with the fast smooth particle mesh Ewald method with a cut off of 12 Å [74,75]. Bonds involving hydrogen atoms were constrained with the LINCS algorithm [76,77]. To improve the sampling, we ran replica exchange simulations [78] using six replicas at closely spaced temperatures (303.15–303.20 K). To further increase the sampling, we ran two rounds of replica exchange. In reA, all replicas were initiated from the starting model (reA1-6) and in reB the replicas started from reA frames where the N-terminal tail was in extended positions (S1 Fig). Both reA and reB replicas showed RMSD equilibration of the N-terminal tail after ~50 ns, with some variation between replicas (S10 Fig).

## Computational analyses

Given the size of the system, we used a replica exchange protocol to enhance the sampling rather than attempting to determine absolute values of thermodynamical properties of the system, such as specific heat. Thus, for each condition (with and without $Cu^+$) we identify statistically significant structural differences based on 12 simulations starting either from different random seeds or from different extended conformations. For all analyses, the domains and linkers were defined as: N-terminal peptide (5–56), MBD1 (57–125), MBD1-MBD2 linker (126–142), MBD2 (143–210), MBD2-MBD3 linker (211–256), MBD3 (257–327), MBD3-MBD4 linker (328–358), MBD4 (359–426), MBD4-MBD5 linker (427–487), MBD5 (488–555), MBD5-MBD6 linker (556–563), MBD6 (564–631), MBD6-M linker (632–642), M domain (643–785 + 915–1002 + 1305–1378), A domain (786–914), P domain (1003–1034 + 1195–1304) and N domain (1035–1194). For RMSD calculations and alignments, the loop regions of the M and N domains were excluded. Simulation analyses were performed with tools from the GROMACS 2019 software package [66] and the VMD 1.9.3 software package [79]. Statistical analyses were performed with the python scipy.stats and scikit-learn packages. Plots and molecular graphics were produced with python matplotlib, VMD 1.9.3 [79] and PyMOL 2.3.0 [80]. Residue-residue contacts were calculated using tcl scripts in VMD. Contacts were defined as non-hydrogen atoms from separate residues within 4.5 Å of each other and contacts were calculated across all frames. Contacts within domains or linkers were not included.

Domain-domain center-of-mass distances were calculated using the gromacs module gmx distance. To test for differences between the apo and holo simulations, we performed independent t-tests for all two-domain combinations, using the trajectory average distance from each replica as samples (12 apo/holo samples). To test for differences in domain positions between the apo and holo simulations, we performed independent t-tests on the x, y and z coordinates of the center-of-mass of all domains calculated with the gromacs module gmx traj. All trajectories were aligned by the M and P domains. Independent t-tests were performed after checking for equal variance with Levene's test ($p < 0.05$). Trajectory averages from each replica were used as samples (12 apo/holo samples). Cross-correlations were calculated from replica averages of all 24 trajectories.

## Contact predictions by evolutionary contacts

Contact predictions were made using the EVcouplings [53] and RaptorX [81] servers. EVcouplings was run with ATP7B residues 481–1180 (S11 Fig) with a bitscore of 0.7. This resulted in a multiple sequence alignment that included coverage of MBD6 as well as the core domains. For the RaptorX analysis, 1260 sequences annotated to be copper transporting ATPases (with CPC, DKTGT, YN and MXXXS motifs) were aligned using the ClustalO algorithm with default parameters in Jalview [82]. After aligning the sequences, all sequences shorter than 1100 residues were discarded to ensure that only sequences with multiple MBDs remained. The remaining sequences (97 sequences) were trimmed to accommodate the RaptorX residue limit of 1100 residues. Thus, amino acids 1–49 (N-terminal tail upstream MBD1), 640–776 (between MBD6 and the A-domain), 922–994 (between the A-domain and P-domain) and 1313–1465 (C-terminal tail) were omitted from the alignment in the RaptorX analysis. The resulting contact map (S8 Fig) shows predicted evolutionary pairs (black dots) within a $C_\beta$-$C_\beta$ distance of 8 Å. Darker color indicates higher probability.

## Supporting information

**S1 Fig. Free energy surfaces from individual replica exchange runs.** Free energy surfaces from the reA_apo (**A**), reA_holo (**B**), reB_apo (**C**) and reB_holo (**D**) simulations, defined by

radius of gyration and RMSD to the N-terminal tail in the starting structure. Starting points for the reB simulations are marked with blue dots.
(TIF)

**S2 Fig. Cross-correlations between N-terminal tail radius of gyration and MBD positions.** Cross-correlation matrix showing cross-correlations between the x, y and z positions of the MBDs and the radius of gyration of MBD1-3, MBD1-4 and MBD1-6. Red shows a positive correlation and blue means a negative correlation. The upper diagonal shows all correlations and the lower diagonal shows correlations with $p < 0.05$.
(TIF)

**S3 Fig. MBDs interact extensively with inter-MBD linkers. (A)** Residue-residue contacts for MBD1-4 for holo (red) and apo (blue), with contacts between MBDs marked with purple boxes. Contact frequencies between residues in **(B)** MBD1 and the MBD1—MBD2 linker, **(C)** MBD2 and the MBD1-MBD2 linker, **(D)** MBD2 and the MBD2-MBD3 linker, and **(E)** MBD3 and the MBD2-MBD3 linker.
(TIF)

**S4 Fig. Contact surfaces between MBD2 and MBD3. (A)** Contact frequency per residue in MBD2 towards MBD3 and in **(B)** MBD3 towards MBD2. **(C)** Cartoon representation of MBD2 and MBD3 in the apo interaction position, with high contact frequency residues in blue (apo) and red (holo). The copper-binding cysteines are shown as sticks. **(D)** Cartoon representation of the holo interaction position, following the representation style used in (C), with copper shown as orange spheres.
(TIF)

**S5 Fig. AlphaFold model of ATP7B.** The M domain is shown in brown, the A domain in yellow, the P domain in blue, the N domain in red and the MBDs in green. **(A)** Overall view of the AlphaFold model, covering MBD5, MBD6 and the core domains. **(B)** Close-up view of the MBD5 copper-binding site and the copper-accepting Met729 in the membrane domain copper binding site, with cys-met sulfur distances in Å. **(C)** Alignment (by M and P domains) of the AlphaFold model with a representative structure from the holo simulations. The AlphaFold model is shown in lighter colors. **(D)** Alternate view of C), MBD6 not shown.
(TIF)

**S6 Fig. Co-evolution predicted position of MBD6. (A)** Residue-residue contacts predicted by the EVcouplings server. Predicted contacts with a probability $> 0.99$ are shown as red dots, with contacts in the simulation starting structure in gray. **(B)** Position of MBD6 based on the coevolution contacts with MBD6 (black), Asn581-Ser726 (red), Ser602-Lys785 (yellow), Lys603-Glu904 (green), and Leu605-Glu905 (cyan). The rest of the protein is shown in gray.
(TIF)

**S7 Fig. Visualisation of contacts between the A domain and MBD5. (A)** MBD5 contact frequency against the A domain, and **(B)** A domain contact frequency against MBD5. **(C)** High contact frequency residues in the A domain are shown in magenta (cluster one, Thr801, Glu802), cyan (cluster two, residues in the range 810–826) and red (cluster 3, residues in the range 892–908), with the rest of the A domain in yellow. **(D)** High contact frequency residues in MBD5 are colored cyan (lower frequency) and blue (higher frequency), with the rest of MBD5 in green. **(E)** The A domain is shown in yellow and MBD5 is shown in green, with high contact frequency residues colored as in C) and D). The M domain is shown in brown and the P domain in blue.
(TIF)

**S8 Fig. Evolutionary covariation analysis of ATP7B.** Analysis of evolutionary coupled amino acids was performed using the RaptorX server. Residues 1–49, 640–776, 922–994, 1313–1465 were omitted in the analysis (see Methods section). The resulting contact map shows predicted evolutionary pairs (black dots) within a $C_\beta$-$C_\beta$ distance of 8 Å. Darker color indicates higher probability with the strongest identified interaction being in-between MBDs (red boxes).
(TIF)

**S9 Fig. Copper coordination geometry. (A)** Copper-sulfur distances for all copper-binding cysteines in the reA_holo simulations. **(B)** Sulfur-copper-sulfur angles for all MBDs in the reA_holo simulations.
(TIF)

**S10 Fig. N-terminal tail RMSD.** RMSD of the N-terminal domain and fitted logarithmic functions for **A)** reA_holo, **C)** reB_holo, **E)** reA_apo and **G)** reB_apo. Average derivative of the fitted logarithmic functions for **B)** reA_holo, **D)** reB_holo, **F)** reA_apo and **H)** reB_apo as solid black lines, with +/- standard deviation as dotted lines.
(TIF)

**S11 Fig. Residues included in the coevolution analysis.** Residues that were not included are colored gray, the others are colored brown (M domain), yellow (A domain), blue (P domain), red (N domain), and green (MBD5 and MBD6).
(TIF)

## Acknowledgments

Computational resources were provided by the Swedish National Infrastructure for Computing (SNIC) through the High-Performance Computing Center North (HPC2N) under projects SNIC2022/23-211, SNIC2021/5-301, SNIC2021/23-23.

## Author Contributions

**Conceptualization:** Fredrik Orädd, Magnus Andersson.

**Formal analysis:** Fredrik Orädd.

**Funding acquisition:** Magnus Andersson.

**Investigation:** Fredrik Orädd, Jonas Hyld Steffen, Pontus Gourdon, Magnus Andersson.

**Methodology:** Fredrik Orädd, Magnus Andersson.

**Project administration:** Magnus Andersson.

**Resources:** Fredrik Orädd, Magnus Andersson.

**Supervision:** Pontus Gourdon, Magnus Andersson.

**Validation:** Fredrik Orädd.

**Visualization:** Fredrik Orädd, Jonas Hyld Steffen.

**Writing – original draft:** Fredrik Orädd, Jonas Hyld Steffen, Pontus Gourdon, Magnus Andersson.

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
