## [Decision Letter · Decision Letter 0]

18 May 2022

Dear Mr Orädd,

Thank you very much for submitting your manuscript "Copper binding leads to increased dynamics in the regulatory N-terminal domain of full-length human copper transporter ATP7B" for consideration at PLOS Computational Biology.

As with all papers reviewed by the journal, your manuscript was reviewed by members of the editorial board and by three independent reviewers. In light of the reviews (below this email), we would like to invite the resubmission of a significantly-revised version that takes into account the reviewers' comments.

We cannot make any decision about publication until we have seen the revised manuscript and your response to the reviewers' comments. Your revised manuscript is also likely to be sent to reviewers for further evaluation.

Sincerely,

Turkan Haliloglu

Associate Editor

PLOS Computational Biology

Nir Ben-Tal

Deputy Editor

PLOS Computational Biology

Reviewer's Responses to Questions

**Comments to the Authors:**

Reviewer #1: In this manuscript, MD simulations have been performed for the first time on the full N-terminal domain and ATP7B core domain. This is a very large system, and the conclusions drawn from this study are highly important, especially since till now no experiments have been performed on the whole ATP7B protein.

I therefore support publication, I have few minor suggestions:

1. In the first paragraph of the results section, it will be nice if the authors can better explain how the homology model was created. Although it is explained in the methods, maybe also mention it here in brief, it will be easier to follow.

2. lines 182-183, I suggest to add the free energy values.

3. There is some experimental work that have been done on MBD1-3, MBD3-4, and MBD4-6, as a function of Atox1 and copper binding that support the conclusions of this manuscript, that was not mentioned here.

Reviewer #2: The article entitled ‘ Copper binding leads to increased dynamics in the regulatory N-terminal domain of full length human copper transported ATP7B’ reports an extensive simulation study based on brute force MD and replica exchange simulations.

The simulations, based on an extensive modeling of the full length structure, show the influence of copper binding on the dynamics of the MBDs and suggest a possible mechanism of copper delivery from MBD5 to the transport region.

The article is interesting and has merit, but I think the authors need to clarify some points.

In general, I think the main issue is that not many details about the replica exchange simulations are provided. 200 ns per replica for a structure of this size is sure not enough and no details are provided on the analysis of the convergence of the results. This is a very important point to understand he reliability of the results

In addition, the free energies for individual replica exchange runs of the each holo and apo systems seem to be very different.

Many computational studies have been done to understand the effect of copper binding on the MBD and Atox1. These should be properly acknowledge. See for example Qasem Metallomics 2019 11 (7), 1288-1297, Magistrato Current Opinion in Structural Biology 2019 58, 26-33

In addition, since the coordination geometry of Cu may be relevant to understand e the structural functional and dynamics properties of the systems I think the author should explicitly state which are the parameters used and the geometry. How does the Cu coordination compare with B. T. Op’t Holt Biochemistry, 2007, 46, 8816–8826 and to Qasem Metallomics 2019 11 (7), 1288-1297?

Reviewer #3: The authors describe a molecular dynamics simulation of the chain of the cytosolic metal binding domains (MBDs) in the copper transporter ATP7B in the context of the whole protein. They show that in the copper-bound form, MBD2 and MBD3 experience greater freedom of motion resulting in a less compact arrangement of the MBD chain. This is consistent with a previously published model based on the NMR studies of the MBD1-6 fragment, which predicts transition of the MBD1-3 domain cluster to a more open conformation, characterized by higher domain mobility, following copper transfer from the copper chaperone ATOX1 to the MBDs. MD simulations in the present work also show repositioning of MBD5, which is directly implicated in copper transfer to the copper-binding site in the transmembrane domain, in the holo-form of ATP7B.

While the general agreement between the computer simulation and the mechanistic models based on the previously published experimental data is important, the true value of molecular dynamics simulations is in providing non-trivial specific predictions that can be further tested experimentally, most commonly by site-directed mutagenesis combined with functional assays. This is where the paper falls a little short.

The authors identified some persistent interdomain residue contacts between the A-domain and MBD5 that show a change between the apo- and holo-forms, but there is very little specifics about the MBD1-3 group. One important question that remains unanswered is which interdomain residue contacts stabilize a more compact arrangement of the MBD1-3 domain cluster in the apo-form, and whether these contacts are disrupted by the copper binding in the simulation. As suggested by the authors, there may be more than one set of such contacts, considering the highly dynamic domain arrangement. Yet, SAXS (Yu et al, 2017, ref. 32) shows a distinct shape of the conformational space occupied by the MBD1-3 cluster in the apo-form, indicating well-defined domain-domain interactions. Perhaps, conducting MD simulation with copper bound only to the MBDs 1-3, as opposed to all six MBDs, will provide some additional insight.

**Have the authors made all data and (if applicable) computational code underlying the findings in their manuscript fully available?**

Reviewer #1: Yes

Reviewer #2: **No: **details on the calculations are missing (see review)

Reviewer #3: **No: **I was not able to access MD simulation trajectories at the DOI provided

PLOS authors have the option to publish the peer review history of their article (what does this mean?). If published, this will include your full peer review and any attached files.

Reviewer #1: **Yes: **Sharon Ruthstein

Reviewer #2: No

Reviewer #3: No
---

## [Decision Letter · Decision Letter 1]

17 Aug 2022

Dear Mr Orädd,

We are pleased to inform you that your manuscript 'Copper binding leads to increased dynamics in the regulatory N-terminal domain of full-length human copper transporter ATP7B' has been provisionally accepted for publication in PLOS Computational Biology.

Best regards,

Turkan Haliloglu

Academic Editor

PLOS Computational Biology

Nir Ben-Tal

Section Editor

PLOS Computational Biology

Reviewer's Responses to Questions

**Comments to the Authors:**

Reviewer #1: I have no more comments

Reviewer #3: The authors have adequately addressed all the comments.

**Have the authors made all data and (if applicable) computational code underlying the findings in their manuscript fully available?**

Reviewer #1: Yes

Reviewer #3: Yes

PLOS authors have the option to publish the peer review history of their article (what does this mean?). If published, this will include your full peer review and any attached files.

Reviewer #1: No

Reviewer #3: No

---

## [Editor Report · Acceptance letter]

1 Sep 2022

PCOMPBIOL-D-22-00531R1 

Copper binding leads to increased dynamics in the regulatory N-terminal domain of full-length human copper transporter ATP7B

Dear Dr Orädd,

I am pleased to inform you that your manuscript has been formally accepted for publication in PLOS Computational Biology. Your manuscript is now with our production department and you will be notified of the publication date in due course.

With kind regards,

Zsofia Freund
